# Understanding the Impact of Contemporary Racism on the Mental Health of Middle Class Black Americans

**DOI:** 10.3390/ijerph20031660

**Published:** 2023-01-17

**Authors:** Darrell Hudson, Akilah Collins-Anderson, William Hutson

**Affiliations:** Brown School, Washington University in St. Louis, St. Louis, MO 63130, USA

**Keywords:** depression, Black Americans, racism, socioeconomic status, discrimination, vigilance

## Abstract

Evidence from previous research indicates that while socioeconomic status (SES) narrows Black-White health inequities, these inequities do not completely disappear, and in some cases, worsen. Why do Black-White health inequities persist, even when controlling for SES? It is critical to examine how perceptions of unfair treatment, especially those that are nuanced and subtle, affect the mental health of Black Americans with greater levels of SES. This study, using a new sample composed exclusively of college-educated Black Americans, investigated whether experiences related to racism were associated with poorer mental health. Qualtrics provided the sample from their nationwide panelists that met the research criteria. Inclusion criteria included the following: (1) self-identified as Black or African American; (2) at least 24 years old; (3) completed a 4-year college degree or higher. The findings from this study indicated that the effects of unfair treatment are significantly associated with poorer mental health. These findings highlight the insidious nature of contemporary racism as the everyday experiences of unfair treatment have a tremendous effect on depressive symptoms among this sample of college-educated Black Americans. Efforts to simply improve SES among historically marginalized groups will not bring about health equity. Findings from this study indicate that there are mental health costs associated with upward social mobility. It is likely that these costs, particularly the experience of everyday unfair treatment, likely diminish the social, economic and health returns on the human capital.

## 1. Introduction

A common assumption among biomedical and public health researchers is that Black-White health inequities in the United States can be reduced or eradicated by narrowing the gap in socioeconomic status (SES) between Black and White Americans [1,2,3,4,5,6,7,8]. Although SES is a fundamental cause of health, SES is inextricable from racist policies and practices that have led to tremendous inequities in SES and fuel contemporary health inequities. Therefore, it is not surprising that Black-White health inequities stubbornly persist, even when accounting for adult SES [4,9]. Evidence from previous research indicates that while SES narrows racial disparities in health, these inequities do not completely disappear [1,2,3,4]. Findings from previous research have indicated that Black-White health inequities not only persist when there are similar levels of SES between Black and White Americans, but these disparities worsen [5,6]. For example, researchers have found that Black Americans with greater levels of SES have worse health outcomes, particularly birth-related outcomes such as pre-term birth and infant mortality, compared to White Americans who have lower levels of SES [7,8]. Johnson found that White high school graduates live longer than Black Americans who have completed college or have graduate or professional degrees [9]. These findings are alarming, yet the exploration of factors that could explain why Black-White health inequities persist when accounting for the effects of SES is understudied. 

Among the scholarship in the United States that has investigated why Black-White health inequities are not fully explained by SES, researchers have highlighted several key factors including historical racial residential and diminished returns on human capital investments due to exposure to racism [10,11,12]. The goal of this study was to investigate whether exposure to different types of racism is associated with poorer mental health, operationalized as depressive symptoms in this study. Considering that depression is the leading cause of disability, globally, linked to several chronic diseases, such as diabetes and cardiovascular disease, and Black Americans have been historically underserved by behavior healthcare, it is critical to gain a better understanding of how social factors such as perceptions of racism affect depressive symptoms among this population [13,14,15,16,17]. An innovation of this study is that it uses data exclusively comprised of college-educated Black Americans. Moreover, it is important to examine the within-group diversity of Black Americans [18].

### 1.1. Racial Residential Segregation

Due to historical redlining practices, residential steering, highway construction and other policies and practices rooted in racism, the United States remains hyper-segregated [19,20,21,22]. Historical racial residential segregation fuels contemporary inequities in neighborhood quality and access to health care among other critical health-promoting resources including but not limited to educational quality and healthcare for many Black Americans [22]. Not only does deeply entrenched racial segregation restrict available resources, some scholars have argued that it also reinforces negative stereotypes held by White Americans [23]. Simply put, segregation allows for many White Americans go about their lives without direct knowledge of the lived experience of Black Americans, including contemporary manifestations of historical legacies of racism experienced by Black Americans [23,24]. Indeed, scholars have noted that the conspicuous absence of Black Americans is considered an indicator of a good school district or neighborhood [23,25]. Other researchers have observed that White Americans are more likely to rate all White neighborhoods as more desirable compared to those that are more integrated [26]. These factors contribute to biases and stereotypes held by White Americans, especially those embedded in the media [27,28].

### 1.2. Diminished Returns

Due to pervasive, deeply entrenched levels of racial residential segregation throughout the United States, Black Americans often enter predominantly White spaces in order to pursue educational and occupational opportunities [22,28,29]. This means that Black Americans often leave their neighborhood and network origins, often navigating environments that are not only different than the ones they were accustomed to, but sometimes encountering racism and poor treatment [12,29]. Similarly, as Black Americans with greater levels of SES seek out more favorable residential and consumer environments, they often enter predominately White spaces [30]. Findings from prior research indicate that Black Americans with greater levels of SES are exposed to more racism and the effects of this exposure undermine the health benefits associated with greater levels of SES [10,11,31]. Exposure to racism at multiple levels is posited to be a major contributing factor fueling Black-White health inequities. Perhaps unsurprisingly, evidence from qualitative studies has documented the stressors that middle-class Black Americans experience which include direct, overt experiences of differential treatment. In addition to these overt experiences of unfair treatment, contemporary manifestations of racism are often considered to be subtle and nuanced, sometimes termed microagressions or manifestations of implicit bias [12]. 

Scholars have argued that in the post-civil rights era, there is an “ordinariness” of racism present in society in which structural inequities are present but there is not a dominant narrative to link these inequities to racism [32]. For example, Bonilla-Silva argues that contemporary racism is often “colorblind” [33]. Bonilla-Silva has argued that notwithstanding members of extreme White supremacist organizations, most White Americans do not consider themselves to be racist and have developed strategies to highlight non-racial explanations, such as market dynamics or cultural deficits among Black Americans, to explain contemporary Black-White inequities [33]. 

Black Americans navigating predominately White spaces are often vigilant against instances of unfair treatment, sometimes using strategies that Black Americans may use to mitigate potential poor treatment. For example, findings from qualitative studies indicate that Black Americans augment their style of dress, hair and diction in an attempt to gain respectability and guard against unfair treatment when navigating predominately White settings [12,30,34,35]. Vigilance has been characterized as anticipatory stress and has been shown to be deleterious to several health-related outcomes including sleep and BMI [27]. 

Findings from previous research have indicated how difficult it is to accurately measure discrimination, especially within the modern context of “colorblind” racism [33]. As such, there are nuanced instances that Black Americans experience, particularly when navigating predominately White spaces [11,12,35,36]. These spaces are often educational or workplace settings, especially for Black Americans with greater levels of SES. There is a need to investigate how more subtle forms of racism, which may be perceived as unfair treatment. Similarly, it is critical to investigate how Black Americans may ruminate on previous experiences of unfair treatment or substantially augment their public presentations of self in order to prevent unfair treatment. 

### 1.3. Study Rationale

Experiences of unfair treatment can also influence expectations for future experiences of negative treatment in similar settings or situations. It is critical examine how perceptions of unfair treatment, especially those that are nuanced and subtle, affect the mental health of Black Americans with greater levels of SES. This study, using a new contemporary sample composed exclusively of college-educated Black Americans, investigated whether experiences related to racism were associated with poorer mental health. Gaining insights related to the experiences of contemporary racism on mental health may help provide guidance to improve future research related to the study of racism and health in addition to aspects of contemporary life that practitioners should take note. 

## 2. Materials and Methods

### 2.1. Data Collection

Panel aggregator, Qualtrics, provided the sample from their nationwide panelists that met the research criteria. We used the Qualtrics survey tool to design and distribute the online survey used to design the survey. In addition to the use of Qualtrics as a survey tool, the Qualtrics XM Platform Research Services team provided consultation services on the survey design and sampling methodologies, and distribution, data collection and quality control of survey responses via Qualtrics’ panels. Qualtrics panel recruitment is derived from traditional active market research groups and social media. Inclusion criteria included the following: (1) self-identified as Black or African American; (2) at least 24 years old; (3) completed a 4-year college degree or higher. Qualtrics sent eligible panel members an email invitation or prompt to participate in the online survey. Potential respondents were sent an email invitation explaining the purpose of the study, the estimated length of time to complete the survey, and what participation incentives were available. The average time of survey completion was 25 min. Financial incentives were distributed directly to participants through Qualtrics. Participant consent was implied when the respondent clicked on the hyperlink to the survey. The landing page of the survey included an additional copy of the study overview and informed consent notification. Our final total sample included 526 respondents drawn from across the United States. 

The study was conducted according to the guidelines of the Declaration of Helsinki, and approved by the Institutional Review Board of Washington University in St. Louis. Informed consent was obtained from all participants involved in the study.

### 2.2. Measurement

Depressive symptoms were assessed using the Patient Health Questionnaire 9-item version (PHQ-9), a validated screener for depression which has been found to function similarly across different racial and ethnic groups [37,38]. The self-report questionnaire consists of the nine Diagnostic and Statistical Manual depressive disorder criteria such as anhedonia, depressed mood and trouble sleeping experienced over the last two weeks. In line with current research and clinical practice, patients who scored ≥10 were classified as meeting criteria for depression care [37,39]. We also examined severity of depressive symptoms using validated cut points for depression severity: moderate [PHQ score 10–14], moderate-severe [PHQ score 15–19] and severe [PHQ score > 19] [37,40] We also examined the PHQ as a continuous outcome given previous concerns about using thresholds and cut points to identify depressive symptoms in previous research [41,42]. 

The survey items included gender, age, participant race/ethnicity and marital status. Extensive SES information, including years of education, household income, and indices of wealth, namely estimates of assets and home ownership status, was collected from all participants. 

Exposure to discrimination was assessed by the major and everyday discrimination scales via online survey [43]. The everyday discrimination scale (10-item version) was scored as the sum of 10 items designed to measure the frequency of routine experiences of unfair treatment [44]. Respondents were asked the following questions: ‘‘In your day-to-day life how often have any of the following things happened to you?’’ The ten domains included the following items: being treated with less courtesy than others receive; receiving less respect than others; receiving poorer service than others in restaurants or stores; people acting as if you are not smart, they are better than you, they are afraid of you, they think you are dishonest; being called names or insulted, being threatened or harassed, and being followed around in stores because of race. 

The major discrimination scale reflects the sum of the unfair events related to the following events: being unfairly fired from a job, unfairly not hired, unfairly denied a job promotion, unfairly denied a bank loan, unfairly discouraged from seeking more education, unfairly stopped by the police, unfairly prevented from moving into a neighborhood, neighbors making life difficult for respondents, and receiving poorer service because of race. 

Vigilance was measured using the heightened vigilance scale (abbreviated four-item version). The scale included the following questions: In your day-to-day life, how often do you do the following things: (1) try to prepare for possible insults from other people before leaving home; (2) feel that you always have to be very careful about your appearance to get good service or avoid being harassed; (3) carefully watch what you say and how you say it; (4) try to avoid certain social situations and places. Frequency responses for the items included the following: almost every day; at least once a week; a few times a month; less than once a year; never.

For descriptive analyses, we estimated means with standard errors of continuous variables and percentages of categorical variables in the total sample. In our multivariate approach, we used linear regression to examine the relationship between SES, perceptions of discrimination and unfair treatment and depressive symptoms. We first examined the relationship between SES and depressive symptoms. We then investigated the bivariate relationship between exposure to racism and depressive symptoms. Next, we added the sociodemographic and SES factors to the model to adjust for the effects for these additional factors. We also used probit regression models to examine the recommended PHQ-9 cutpoints (e.g., none/mild, moderate, moderately severe, severe) to examine the association between depressive symptoms, discrimination and heightened vigilance. 

## 3. Results

The sociodemographic characteristics of the sample are displayed in Table 1. The total sample was 526 participants, all of whom reported that they held at least a four-year college degree and indicated that they self-identified as Black or African Americans. Of this sample, nearly 36% of the sample indicated that they had a graduate or professional degree. The gender distribution in the sample was nearly equal, with 50% of the respondents indicating that they were women. Nearly 40% of the study participants reported that they were married or living with a partner. Approximately 46% of the sample reported that they had never been married. Nearly 34% of the sample reported a household income of more than $80,000 per year with 21.4% reporting household incomes between $40,000–$64,999 and 17.6% reporting incomes between $65,000–$79,999. With regard to financial assets, 67.2% reported assets of $10 k or more. The majority of respondents reported that they were middle class (48.3%), 21% reported that they were upper middle class and 10% reported that they were upper class. 

There were no significant associations between SES-related factors, including education, income, assets and home ownership, and depressive symptoms among this sample. While age was negatively associated with depressive symptoms, there was no significant association between gender and depressive symptoms. Because there were no significant associations between SES and depressive symptoms, we did not display interaction analyses here. There were no significant interactions in analyses. 

As depicted in Table 1, nearly 40% (39.97) of the sample reported symptoms that were indicative of significant depressive symptoms that warrant follow up from a provided according to the PHQ-9 with symptoms categorized as moderate, moderately severe, and severe. Additionally, 15.5% of the sample reported that they have been diagnosed with depression by a provider at some point in their lives. There was a significant negative association between age and depressive symptoms such that older respondents reported fewer depressive symptoms. There were no significant gender or SES associations, including home ownership status and estimates of assets, with depressive symptoms. 

In the bivariate analyses, which are not depicted here, unattributed everyday discrimination was significantly associated with depressive symptoms (b = 0.26; *p* < 0.001), explaining nearly 22% of the variance in depressive symptoms among this sample of Black middle-class participants. This association remained significant (b = 0.22; *p* < 0.001) once the other sociodemographic factors were added to the model, continuing to explain about 22% of the variance in depressive symptoms. We also examined the association between everyday discrimination and depressive symptoms using the categorical version of the PHQ-9 using the recommended cutpoints. Bivariate analyses revealed that there was a significant association (*p* < 0.001) between all moderate, moderate-severe and severe levels of depressive symptoms and everyday discrimination. As depicted in Table 2, these associations remained significant once SES and other sociodemographic factors were included in the analysis. 

There was a positive association between major discrimination and depressive symptoms (b = 0.59; *p* < 0.001), explaining about 7% of the variance in depressive symptoms. This association remained significant after adjusting for sociodemographic factors including age, gender, and SES-related factors (b = 0.71; *p* > 0.001). Adjusting for the additional covariates explained about 16% of the variance in depressive symptoms. 

Examining the bivariate association between major discrimination and the PHQ-9 cutpoints, depicted in Table 2, analyses revealed a significant association between moderate depressive symptoms and major discrimination (*p* = 0.006) and severe depressive symptoms (*p* = 0.02). After adjustment for pertinent SES and sociodemographic factors, there was a significant association between major discrimination and all the PHQ categories (moderate *p* < 0.001); moderately severe *p* = 0.012; severe *p* = 0.004).

There was a significant association between the vigilance measure and depressive symptoms (b = 0.33; *p* < 0.001), explaining about 6% of the variance in depressive symptoms. This association remained significant once the covariates were adjusted for in the model (b = 0.305; *p* < 0.001) and nearly 18% of the variance in depressive symptoms was explained once these factors were added. Bivariate analyses for the categorical analyses indicated there was a significant association between vigilance and depressive symptoms but only for the severe category (*p* < 0.001). This pattern of association remained the same once SES and sociodemographic factors were adjusted for in the model as depicted in Table 3. 

## 4. Discussion

The findings from this study, which were drawn from a sample of college-educated Black Americans, indicated that there is a significant association between everyday discrimination and depressive symptoms. Everyday discrimination alone accounted for 22% of the variance in depressive symptoms. Similarly, reports of major discrimination were significantly associated with more depressive symptoms. As stated, the effects of SES did not diminish the potency of discrimination. Results from prior studies, however, have found that greater levels of discrimination undermine the health benefits of greater SES levels. 

The findings from this study also lend support to the diminishing returns hypothesis, which posits that Black Americans do not experience the same health returns as their White SES counterparts due to exposure to racism [10,11]. The findings from this study provide additional evidence that exposure to discrimination is quite prevalent among middle-class Black Americans as prior research findings have indicated that Black Americans with greater levels of income and education experience greater levels of discrimination than working class or poorer Black Americans [10,31]. These findings corroborate with results from previous research that has documented the pernicious health effects of discrimination on mental health among Black Americans [10,31,45,46]. Additionally, the findings provide additional evidence that exposure to discrimination is quite prevalent among middle-class Black Americans. 

Scholars have argued that while overt forms of racism may be less common in contemporary times, the health impact of unfair treatment is still significant. One of the innovations of this study was the inclusion of a measure of heightened vigilance, which is often considered related to anticipatory stress based on expectations formed in previous encounters, through socialization (e.g., messages received from social networks), as well as hearing about vicarious experiences of unfair treatment among members of their social network and beyond (e.g., vis-a-vis social media) [47,48]. 

Several limitations should be considered when interpreting the results of this study. There are limitations to using panel aggregators such as Qualtrics [49,50]. Notably, they may not capture specific hard-to-reach populations that have limited internet access and the need to provide specific quota/detailed inclusion/exclusion criteria may inadvertently exclude a sub-sample of respondents [50,51]. However, Qualtrics quality-control checks through attention filters and data review processes are powerful tools for assessing the accuracy of survey data. Qualtrics panel recruitment compared to traditional (clinical/community) recruitment methods has several advantages including lower relative cost, faster recruitment due to decreased barriers for study participation, increased confidentiality and diverse samples [50]. Another limitation is that the investigation of unfair treatment was at the individual/interpersonal levels. It is important to consider multiple levels of influence with regard to unfair treatment, especially considering the spatial components and racial composition aspects of this work. Additionally, future research is needed to evaluate the effects of vicarious experiences of unfair treatment, especially as incidents that occur in public settings are widely disseminated via social media, entering the consciousness of many, not just those immediately affected. Relatedly, another limitation is that we limited the findings of this study to unattributed discrimination. Specifically, we did not display results for perceptions of discrimination attributed directly to race. We felt this discussion was most appropriate considering the challenges of attributing differential treatment to only one aspect of an individual’s identity [52,53]. We found that unattributed everyday and major discrimination were more powerful, in relation to depressive symptoms, than race-attributable discrimination. Lastly, this paper focused on the experiences of cisgender Black men and women. While we queried respondents about gender identity (e.g., transgender, nonbinary), there were not respondents that reported that they were transgender or identified as nonbinary in this sample. However, this is a limitation of the study fundings as there are Black people with gender identities that are non-binary and it is critical to examine their experiences of discrimination, as it is likely that they face harmful gendered hostility in addition to facing racial discrimination. 

## 5. Conclusions

Black-White health inequities narrow but are not eliminated with adjustment for SES. As the findings from this study indicate, drawing from a sample of Black Americans with greater levels of SES, the effects of unfair treatment are related to poorer mental health. These findings highlight the insidious nature of contemporary racism as the everyday experiences of unfair treatment have a tremendous effect on depressive symptoms among this sample of college-educated Black Americans. Due to the difficulty of attributing perceptions of unfair treatment to the intersecting identities that Black Americans have as well as the heightened vigilance against the possibility of unfair treatment, it is critical to examine how these factors affects depression and other health outcomes. 

Efforts to simply improve SES among historically marginalized groups will not bring about health equity. Racism determines which neighborhoods are seen as most valuable as well as access to key health resources. Furthermore, findings from this study as well as those from previous research indicate that there are health costs associated with upward social mobility, including constant navigation and negotiations with people in predominantly White spaces. It is likely that these costs, particularly the experience of everyday unfair treatment, likely diminish the social, economic and health returns on the human capital investments that people of color make. Therefore, greater efforts are needed to understand the unique experiences Black Americans that could undermine their investments in human capital. 

## Figures and Tables

**Table 1 ijerph-20-01660-t001:** Descriptive and Sample Characteristics Statistics (N = 528).

	M or %	SD or N	Range (If Applicable)
Age	38.9	12.5	18–66
Gender (ref Male)	48.8	258	1–3
Education			
Bachelor’s Degree *	64.5	341	
Professional/Graduate degree	35.4	187	
Relationship Status			
Married/Partnered *	39.6	209	
Divorced/separated/widowed or never married	60.4	319	
Annual income			
Less than $10,000 *	4.3	23	
$10,000–$25,999	9.2	49	
$26,000–$39,999	10.9	58	
$40,000–$64,999	21.4	113	
$65,000–$79,999	17.6	93	
More than $80,000	33.5	177	
Assets			
Less than $10,000 *	25.5	135	
More than $10,000	67.2	355	
Home Ownership			
Rent *	32.7	173	
Own	59.4	314	
Live rent-free	4.5	24	
Depression Diagnosis (ref No)	84.4	446	
Severity of depressive symptoms			
None/Mild *	60.0	317	
Moderate	19.1	101	
Moderately Severe	13.6	72	
Severe	7.2	38	
Patient Health Questionnaire (PHQ-9)	8.3	7.0	0–27
Everyday Discrimination	19.9	12.9	0–50
Major Discrimination	2.1	1.9	0–9

Totals may not add to 528. * Reference group.

**Table 2 ijerph-20-01660-t002:** Multinomial Regression Predicting Severity of Depressive Symptoms.

	Coefficient (95% CI)					
	Moderate		Moderately Severe		Severe	
Age	−0.06 *	(−0.07–−0.43)	−0.05 *	(−0.07–−0.03)	−0.04 *	(−0.06–−0.02)
Gender	−0.08	(−0.43–0.26)	−0.14	(−0.52–0.22)	0.06	(−0.36–0.49)
Education	0.12	(−0.25–0.50)	−0.06	(−0.48–0.34)	0.12	(−0.03–0.58)
Relationship Status	−0.02	(−0.12–0.06)	0.04	(−0.06–0.15)	0.00	(−0.11–0.12)
Annual income	−0.01	(−0.02–0.00)	−0.00	(−0.01–0.10)	−0.02	(−0.06–0.01)
Assets	0.00	(−0.00–0.00)	−0.01	(−0.02–0.00)	0.00	(−0.00–0.01)
Home ownership	0.00	(−0.01–0.01)	0.00	(−0.00–0.02)	0.00	(−0.01–0.02)
Discrimination						
Everyday Discrimination	0.06 *	(0.05–0.08)	0.05 *	(0.03–0.06)	0.66 *	(0.04–0.08)
Major Discrimination	0.12 *	(0.03–0.20)	0.07	(−0.01–0.17)	0.12 *	(0.01–0.22)

* Statistically significant at *p* < 0.05.

**Table 3 ijerph-20-01660-t003:** Association between Hypervigilance and Severity of Depressive Symptoms.

	Coefficient (95% CI)					
	Moderate		Moderately Severe		Severe	
Age	−0.06 *	(−0.07–−0.04)	−0.05 *	(−0.07–−0.03)	−0.05 *	(−0.07–−0.02)
Gender	−0.07	(−0.43–0.27)	−0.13	(−0.51–0.23)	0.03	(−0.40–0.48)
Education	0.09	(−0.28–0.47)	−0.09	(−0.51–0.31)	0.01	(−0.46–0.49)
Relationship Status	−0.03	(−0.13–0.06)	0.04	(−0.06–0.14)	0.00	(−0.12–0.12)
Annual income	−0.01	(−0.02–0.00)	−0.00	(−0.01–0.10)	−0.02	(−0.06–0.01)
Assets	0.00	(−0.00–0.00)	−0.01 *	(−0.02–−0.00)	0.00	(−0.00–0.01)
Home ownership	0.00	(−0.00–0.01)	0.01	(−0.00–0.02)	0.01	(−0.00–0.02)
Hypervigilance	0.03	(−0.00–0.07)	0.03	(−0.00–0.07)	0.10 *	(0.05–0.15)

* Statistically significant at *p* < 0.05.

## Data Availability

The data presented in this study are available on request from the corresponding author.

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
