# Peer review of "Understanding the Impact of Contemporary Racism on the Mental Health of Middle Class Black Americans"

_ijerph, 2023, doi:10.3390/ijerph20031660_

Round 1

Reviewer 1 Report

This is an excellent article. Well grounded and asking an importnat question. I have a couple minor suggestions that could strengthen the article, but I think this is a very sound piece of work as it is. I especially appreciate how it brings a nuanced understanding to health and racialized experiences and how they systematically address any questions. 

Suggestion 1) In the openine paragraph would be good to include something that speaks to this expectation that inquities would dissappear if SES gaps were closed. Just explain why people would even think that SES would erase Racial issues. one citation to substantiate this would be fine. 

Suggestion 2) in the study, you look at women and men, but there are non binary and trasngender Black people. That is not included here and I think there would be a meaningful difference based on this, literature shows that transgender black women for instance experience heightened racialized and gendered hostility. I am aware that this study is not about this, and I think it would be worth acknolwedging this as a limitation and worth integrating in future research. 

Author Response

Reviewer 1:
This is an excellent article. Well-grounded and asking an important question. I have a couple of minor suggestions that could strengthen the article, but I think this is a very sound piece of work as it is. I especially appreciate how it brings a nuanced understanding to health and racialized experiences and how they systematically address any questions. 

We are so appreciative to the reviewer for their very helpful and productive feedback. We believe these suggestions have strengthened the manuscript. Thank you.

Suggestion 1) In the opening paragraph would be good to include something that speaks to this expectation that inequities would disappear if SES gaps were closed. Just explain why people would even think that SES would erase Racial issues. One citation to substantiate this would be fine. 

We agree with this assessment and have added a sentence to the beginning of the first paragraph.

A common assumption among biomedical and public health researchers is that Black-White health inequities in the United States can be reduced or eradicated by narrowing the gap in socioeconomic status (SES) between Black and White Americans ]1-8]. Although SES is a fundamental cause of health, SES is inextricable from racist policies and practices that have led to tremendous inequities in SES and also fuel contemporary health inequities. Therefore, it is not surprising that Black-White health inequities stubbornly persist, even when accounting for adult SES [4,9].

Suggestion 2) in the study, you look at women and men, but there are non-binary and acknowledge people. That is not included here and I think there would be a meaningful difference based on this, literature shows that transgender black women for instance experience heightened racialized and gendered hostility. I am aware that this study is not about this, and I think it would be worth acknowledging this as a limitation and worth integrating in future research. 

We wholeheartedly agree. In our survey, respondents had the option to indicate multiple expressions of gender, including nonbinary and transgender options. However, there were no respondents in our sample that indicated these identities. We did include this as a limitation and attempted to capture your guidance here.

Author Response

We were very disappointed in this review and the critiques are very far off from Reviewer 1 who found the paper to be very well-written. While there were some typos and small errors, the job of a reviewer is not to provide line-by-line grammar edits or in-text citations (the journal has copy editors for that). 

The assertion that the paper was poorly written or needs professional English editing indicates that the reviewer either lacks the expertise to sufficiently review the paper or to access English proficiency.

We do, however, appreciate the reviewer for highlighting the typos and errors. By addressing those, we were able to address and improve the clarity of the manuscript.

Round 2

Reviewer 2 Report

Looks satisfactory.